# The Introduction of the Global Traditional, Complementary, and Integrative Healthcare (TCIH) Research Agenda on Antimicrobial Resistance and Its Added Value to the WHO and the WHO/FAO/UNEP/WOAH 2023 Research Agendas on Antimicrobial Resistance

**DOI:** 10.3390/antibiotics14010102

**Published:** 2025-01-17

**Authors:** Erik W. Baars, Petra Weiermayer, Henrik P. Szőke, Esther T. van der Werf

**Affiliations:** 1Faculty of Healthcare, University of Applied Sciences Leiden, 2333 CK Leiden, The Netherlands; 2Department of Health, Louis Bolk Institute, 3981 AJ Bunnik, The Netherlands; 3WissHom: Scientific Society for Homeopathy, 06366 Koethen, Germany; petra.weiermayer@gmx.at; 4Department of Medicine, University of Witten/Herdecke, 58453 Herdecke, Germany; 5Department of Integrative Medicine, University of Pécs, 7621 Pécs, Hungary; henrik.szoke@etk.pte.hu; 6Homeopathy Research Institute, London SW7 4EF, UK; esthervanderwerf@hri-research.org; 7Bristol Medical School, University of Bristol, Bristol BS8 1UD, UK

**Keywords:** traditional medicine, complementary medicine, medicinal plants, antimicrobial drug resistance, research agenda

## Abstract

**Background/Objectives**: Given the magnitude and urgency of the global antimicrobial resistance (AMR) problem and the insufficiency of strategies to reduce antimicrobial use, there is a need for novel strategies. Traditional, Complementary, and Integrative Healthcare (TCIH) provides strategies and solutions that contribute to reducing (inappropriate) antimicrobial use, preventing or treating infections in both human and veterinary medicine, and may contribute to promoting the health/resilience of humans and animals and reducing AMR. The aims of this study were to present the core results of a global TCIH research agenda for AMR and its added value to two existing global AMR research agendas published in 2023. **Methods**: A survey, interviews, and consensus meetings among network members, as an adapted version of the nominal group technique, were executed to develop the global TCIH research agenda. A comparison of the global TCIH research agenda with the two existing global AMR research agendas was performed. The TCIH additions to these two existing global AMR research agendas were determined. **Results**: The global TCIH research agenda adds to 19 of 40 research priorities of the World Health Organization (WHO) AMR research agenda 2023 and three of the five pillars of the WHO/Food and Agriculture Organization of the United Nations (FAO)/United Nations Environment Programme (UNEP)/World Organisation for Animal Health (WOAH) research agenda 2023. In addition, the TCIH research agenda adds two new research themes with four new research priorities and three new research priorities to already existing themes of the two global AMR research agendas. **Conclusions**: The global TCIH research agenda fits with and adds to two global AMR research agendas and can be used as an additional strategy to reduce AMR and (inappropriate) use of antibiotics.

## 1. Introduction

### 1.1. The Burden of AMR

Resistance to antibiotics and antimicrobials is a complex and growing international public health problem with significant consequences, such as increased mortality and economic impact [1]. Back in 2014, a review on antimicrobial resistance (AMR) from the United Kingdom (UK) stated that “Drug-resistant infections already kill hundreds of thousands a year globally, and by 2050, that figure could be more than 10 million. The economic cost will also be significant, with the world economy being hit by up to US$100 trillion by 2050 if we do not take action” [2].

In the last decade, global, regional, and national strategies were developed to reduce AMR. The most important strategies are (the monitoring of) infection prevention and control of resistant bacteria, research on antibiotic resistance and antibiotic use, appropriate use of antibiotics (e.g., not for viral infections), less antibiotic use (e.g., delayed prescription and alternatives), development of new antibiotics, and understanding the role of and making use of artificial intelligence (AI) in the application of antibiotics [3,4].

However, in human medicine little progress has been made despite all major efforts to reduce antibiotic use and AMR. Between 2000 and 2015, global antibiotic consumption increased by 65% (21.1–34.8 billion defined daily doses (DDDs)). The increase was driven by low-and middle-income countries (LMICs). In high-income countries (HICs), human consumption has increased modestly [5]. In the European Union (EU), between 2016 and 2018, there was only a minimal decrease of 0.4% in the average total (community and hospital sector) consumption of antibacterials for systemic use in humans [6]. In November 2024, the European Centre for Disease Prevention and Control (ECDC) reported that “Between 2019 and 2023, antibiotic consumption in the EU increased by 1%, moving further away from the 2030 target of a 20% reduction recommended by the Council of the European Union” and “While some Member States have made great progress towards their recommended AMR targets, or even in some instances have already reached the recommended targets, the overall picture shows that more specific, intensified interventions are urgently needed across the EU” [7].

For the European Union/European Economic Area (EU/EEA) population-weighted mean consumption of antimicrobials in food-producing animals, expressed in mg/kg estimated of biomass, there was a significant change for the 27 countries included in the analysis. A decline of 32% was observed between 2014 and 2018 in farm animals, while a slight increase was observed in humans [8]. The latest European Surveillance of Veterinary Antimicrobial Consumption (ESVAC) data on sales figures of antimicrobials in food-producing animals show that most of the 25 participating countries have been successful in reducing the use of antimicrobials. Between 2011 and 2022, sales fell by 53% over this period [9].

Nevertheless, the WHO reported alarming resistance rates among prevalent bacterial pathogens in the 2022 Global Antimicrobial Resistance and Use Surveillance System (GLASS). Analyses of the associations between antimicrobial consumption (AMC) and AMR for selected combinations of antimicrobials and bacteria exposed to them demonstrated positive associations between the consumption of certain antimicrobials and resistance to those substances in bacteria from both humans and food-producing animals [10]. The spread of resistant strains of bacteria between humans, animals, and the environment may be triggered through multiple factors and pathways such as antimicrobial exposure, human travel, livestock trade, manure runoff, or water contamination [11,12]. The Central Asian and European Surveillance of Antimicrobial Resistance (CAESAR) network and EARS-Net (European Antimicrobial Resistance Surveillance Network) data from 2023 show that AMR remains a severe threat to public health in Europe [13], that there is still poor progress towards the EU targets on AMR overall and that current public health actions to tackle the situation remain insufficient. In 2022, Murray et al. [14] demonstrated that AMR is a leading cause of death worldwide. The highest burdens are found in low-resource settings, regions where inadequate healthcare resources exist and the healthcare system does not meet acceptable global standards. It is estimated that 4.95 million (3.62–6.57) deaths were associated with bacterial AMR in 2019, including 1.27 million (95% UI 0.911–1.71) deaths attributable to bacterial AMR. At the regional level, the all-age death rate attributable to resistance was highest in western sub-Saharan Africa, with 27.3 deaths per 100,000 (20.9–35.3), and lowest in Australasia, with 6.5 deaths (4.3–9.4) per 100,000. Globally, the WHO GLASS reports the increasing prevalence of AMR with serious therapeutic problems [15].

Regarding the costs of AMR, Hofer [16] estimated that 2.4 million people in Europe, North America, and Australia will die from infections with resistant microorganisms in the next 30 years, with Southern European countries hit the hardest. These infection-related costs could run up to US$3.5 billion per year. Many LMICs already have high resistance rates. For these countries, the expectation is that these rates will increase disproportionately. For example, 40–60% of human infections in Brazil, Indonesia, and Russia are already caused by resistant microorganisms, and resistance is predicted to rise 4–7 times faster in these countries than in other Organisation for Economic Co-operation and Development (OECD) countries. Poudel et al. [17] conducted a systematic review and meta-analysis to demonstrate that the economic burden of antibiotic resistance remains significant.

### 1.2. Global Research Agendas on AMR

In 2023, two global research agendas on AMR were published: the “Global Research Agenda for Antimicrobial Resistance in Human Health” [18] and the “A One Health Priority Research Agenda for Antimicrobial Resistance” [19].

The “Global Research Agenda for Antimicrobial Resistance in Human Health” [18] states: “The goal of this research agenda is to identify and give priority to the research topics with the greatest impact on mitigating antimicrobial resistance in the human health sector, in accordance with objective 2 of the Global Action Plan (i.e., on strengthening the knowledge and evidence base through surveillance and research). The research agenda also aims to foster research by 2030—in accordance with the Sustainable Development Goals timeline—and to catalyse scientific interest and investment among the scientific community and funders on the epidemiology and burden of resistant infections, strategies to prevent infections and the emergence of resistance, how to optimise and best deliver these in low- and middle income countries, and optimised diagnostics and antimicrobial medicines”.

The WHO research agenda contains forty research priorities in thirteen AMR areas across five themes [18] (Figure 1).

The “A One Health Priority Research Agenda for Antimicrobial Resistance” [1] was “designed to further support cross-disciplinary research and to strengthen One Health AMR research capacity and partnerships in low-resource settings”. It states: “The global threat of AMR spreading among humans, animals, plants and the environment necessitates a “One Health” approach in our evermore connected world. One Health acknowledges the connected and interdependent nature of the health of humans, domesticated and wild animals, plants and the wider environment. Research strategies, interventions and policies based on One Health principles are emerging, but require more evidence to understand what works, in which contexts and for whom”.

The research agenda is built on five pillars:**Transmission**: This pillar focuses on where the transmission, circulation, and spread of AMR occur among the environment, plants, animals, and humans, what drives this transmission, where it happens, and its impact. This focus includes transmission dynamics, risk assessment, and modeling, and how practices enacted by humans at the interface between humans, plants, animals, and the wider environment (soil, water, and air) enable the development and spread of resistance.**Integrated surveillance**: This pillar aims to identify priority research questions focusing on cross-sector surveillance that improves common technical understanding and exchange of information. Included are questions about harmonization, effectiveness, implementation of One Health integrated surveillance and applicability to LMICs; it may include considerations for innovative surveillance approaches to AMR.**Interventions**: This pillar covers programs, practices, tools, and activities designed to prevent, contain, or reduce the incidence, prevalence, and dissemination of AMR, including optimal use of existing vaccines and other measures across the One Health spectrum.**Behavioral insights and change**: This pillar focuses on behavioral drivers of AMR by understanding influences on human behavior in different contexts (social influences and support, livelihoods, financial resources, etc.). This pillar operates at multiple levels of complex systems, including organizational structures that enable or disable AMR mitigation, as well as individual and interpersonal sociocultural practices.**Economics and policy**: This pillar addresses investment and action in AMR mitigation from a One Health perspective. Included are policy, governance, legislative and regulatory instruments, cross-sector processes and strategies affecting AMR (e.g., regulation of antimicrobial manufacturing, use, disposal, monitoring), joint planning, and policy goals among ministries. Cost-effectiveness considerations are also included to support the development of the AMR investment case. Finally, this pillar includes financial sustainability and long-term financial impact.

### 1.3. The Scientific Status of Traditional, Complementary, and Integrative Healthcare (TCIH) for the Prevention and Treatment of Infections

The use of TCIH prevention and treatment of infections strategies in human and veterinary medicine is already based on an increasing number of scientific studies and evidence. For example, positive results from systematic reviews on TCIH medicinal products for acute, uncomplicated respiratory tract infections (RTIs) were found for *Andrographis paniculata* [20], *Pelargonium sidoides* for both general and specific upper RTI symptoms (e.g., cough and sore throat) [21,22,23,24], *Echinacea* spp. for the common cold [25,26], and a combination of ivy (*Hedera helix* L.), primrose (*Primula veris* L./*Primula elatior* L.), and thyme (*Thymus vulgaris* L./*Thymus zygis* L.) for coughs only [22]. A systematic review of mechanisms of action [27] demonstrated that *Andrographis paniculata* acts through immunomodulation and antiviral activity, possibly supplemented by antibacterial and antipyretic effects. *Pelargonium sidoides* act through antiviral, indirect antibacterial, immunomodulatory, and expectorant effects. *Echinacea* species likely act through immunomodulation. The combination of ivy/primrose/thyme combines the secretolytic and spasmolytic effects of ivy with the antibacterial effects of thyme.

Several studies examined the effectiveness of homeopathic medicinal products. A recent meta-analyses (MAs) of placebo-controlled randomized efficacy trials of homeopathy for any indication [28] demonstrated “The quality of evidence for positive effects of homeopathy beyond placebo (high/moderate/low/very low) was high for I-HOM [red: individualized homeopathy] and moderate for ALL-HOM and NI-HOM [red: non-individualized homeopathy and all homeopathy types]. There was no support for the alternative hypothesis of no outcome differences between homeopathy and placebo. The available MAs of PRETHAIs [red: placebo-controlled randomized efficacy trials of homeopathy for any indication] reveal significant positive effects of homeopathy beyond placebo. This is in accordance with laboratory experiments showing partially replicable effects of homeopathically potentized preparations in physico-chemical, in vitro, plant-based, and animal-based test systems”. Regarding the treatment of infections, several studies examined the effectiveness of homeopathic medicinal products either in conjunction with antibiotics in the treatment of bacterial infections or with homeopathy as the sole treatment in humans and animals, demonstrating positive results [29]. An economic evaluation of a complex homeopathic medicinal product (MP) used showed that, compared with antibacterial treatment, homeopathy had a significantly higher cure rate in the treatment of acute maxillary sinusitis in adults (11% vs. 59%; *p* < 0.001) at similar or lower costs [30].

An example of an outcome study in the field of anthroposophic medicine is an observational study that studied 529 children <18 years from Europe (Austria, Germany, The Netherlands), the UK, or the United States of America (USA), whose caregivers had chosen to consult physicians offering anthroposophic (A) or conventional (C) treatment for respiratory tract infection (RTI)/otitis media (OM). During the 28-day follow-up, antibiotics were prescribed to 5.5% of A-patients and 25.6% of C-patients; the unadjusted odds ratio for non-prescription in A-patients versus C-patients 6.58 (95%-CI 3.45–12.56); after adjustment for demographics and morbidity 6.33 (3.17–12.64). Antibiotic prescription rates in recent observational studies with similar patients in similar settings ranged from 31.0% to 84.1%. Compared with C-patients, A-patients also had much lower use of analgesics, somewhat quicker symptom resolution, and higher caregiver satisfaction. Adverse drug reactions were infrequent (2.3% in both groups) and not serious [31].

Some TCIH medicinal products are already integrated into conventional guidelines, for example, in Germany: *Thyme/Primrose* and *Pelargonium sidoides* for coughs [32], rhinosinusitis [33], and in the UK: *Pelargonium sidoides* for coughs [34]. All these TCIH medicinal products are registered in Germany with the BfArM (Besondere Therapierichtungen und Traditionelle Arzneimittel) [35] and also have a so-called European Medicines Agency (EMA) status of Traditional use or Well-established use [36]. In the EU, the registration of homeopathic medicinal products without indication and the authorization of homeopathic medicinal products with an indication are laid down in EU Directive 2001/83 [37].

For (recurrent) urinary tract infections ((r)UTIs), a systematic review of TCIH medicinal products demonstrates that the latest published meta-analysis, including 28 trials, reports a clear benefit of *Cranberry* products for preventing (r)UTIs in women. Five TCM formulas were found to be equally or more effective than antibiotics in treating UTIs. Furthermore, *Rosa canina* seems to have the potential to prevent UTIs in women undergoing a cesarean section. ‘Acidif Plus Tablets’ as well as ‘Canephron’ seem to be promising candidates for treating women with uncomplicated rUTIs [38].

High-quality randomized controlled trials (RCTs) supporting the evidence of homeopathy for the treatment of recurrent UTIs are scarce and only available for specific populations [39]; however, a survey on the use and patients’ perceived effectiveness of TCIH and self-care strategies in women with (recurrent) urinary tract infections in the Netherlands showed the homeopath to be the most popular TCIH health practitioner to be consulted for (r)UTIs and the majority of its users perceived the treatment as effective. These findings warrant further studies into the effectiveness of homeopathy in the prevention and treatment of (r)UTIs [40].

In pigs, promising alternatives for the prevention and treatment of infections are probiotics and prebiotics, organic acids such as short-and medium-chain fatty acids, phytogenic substances, bacteriophages, spray-dried plasma, and homeopathy [29,41,42,43]. In chickens, promising alternatives are Egg Yolk Antibodies (EYA), pro- and prebiotics, antimicrobial peptides such as bacteriocins, ß-defensins, protegrins, insect defensins, and homeopathy [44,45,46,47,48,49,50]. In cows, promising TCIH treatments are phytogenic substances, probiotics, bacteriocins, bacteriophages, stem cells, minerals, trace elements, vitamins, short-chain fatty acids, and microbial lipopolysaccharide, and homeopathy [29,51,52].

Several, mostly observational, studies support the hypothesis that doctors who practice TCIH (including and integrating both conventional medicine and complementary medicine) have lower antibiotic prescription rates compared with their conventional colleagues (measured as past use, antibiotics use ever, in the first 12 months of life and after 12 months of life, consumption, prescription rates) and their patient groups have lower antibiotic consumption rates, although in these studies selection bias (e.g., patients who do not want antibiotics may choose a TCIH doctor more often) cannot be ruled out [53,54]. Nevertheless, TCIH practices appear to contribute to a reduction in antibiotic use. A reduction in antibiotic use as a result of TCIH prevention and treatment of infections has also been described for veterinary medicine [55,56].

Overall, there is some evidence that TCIH prevention and some TCIH treatment strategies for infections are effective and safe. Many TCIH treatment strategies for infections are promising, but most lack high-quality evidence [29,53].

### 1.4. Patients’, Animal Owners’, Farmers’ Preferences and Doctors’ (Non-) Prescription of TCIH Medicinal Products

Several surveys demonstrate that many people wish to be treated with TCIH in general [40,57]. Surveys conducted among patients in university hospitals show that more than 50% of patients in oncology, gastroenterology, and even cardiology departments are requesting TCIH treatment and wish to be better informed about it [58,59,60]. The same open attitude towards TCIH can be seen in farmers [55,56,61].

However, there is a complex array of factors that influence the (non-)prescribing of TCIH. The attitudes of both doctors and patients are shown to be of major significance in prescribing decisions. Many doctors do not (want to) prescribe TCIH antibiotic alternatives because of patient pressure to prescribe antibiotics, fear of the ineffectiveness of TCIH treatments, lack of TCIH knowledge in general, insufficient information on the effectiveness and safety of TCIH treatments, (assumed) insufficient regulation of herbal practitioners, concerns about herbal quality control and potential herb–drug interactions, and because of a lack of communication between doctors and patients about this topic [53].

Nevertheless, for example, the field of uncomplicated, acute RTI treatment demonstrates that there is a need and ‘market’ for TCIH. RTIs are among the most common infections experienced in the community and are among the most common reasons for antibiotic prescribing internationally (e.g., [62]). Previous studies show that although antibiotics have small or negligible symptomatic benefits for patients with uncomplicated acute otitis media, pharyngitis, bronchitis, laryngitis, and the common cold, antibiotics are still commonly used for these and other viral respiratory infections (e.g., [63,64]). The same applies to veterinary medicine, as supported by the published figures from 2018, showing 50% of unfounded or improper antibiotic use in veterinary medicine [65]. Effective and safe non-antibiotic TCIH RTI treatment (as an alternative treatment or as part of a delayed prescription strategy) may, therefore, contribute to reducing antibiotic use and prescription and AMR, meeting both doctors’ and patients’ desire for treating RTIs and symptom relief. A systematic review of qualitative studies demonstrates that patients are open to TCIH treatment of acute RTIs but need trusted advice on the safety and effectiveness of TCIH and antibiotics for specific acute RTIs [66]. Trusted advice on the effectiveness and safety of TCIH is needed by farmers and, respectively, the patient’s owners too [61,67,68].

### 1.5. The GIFTS-AMR Project

The Global Initiative for Traditional Solutions to Antimicrobial Resistance (GIFTS-AMR) was a Joint Programming Initiative for Antimicrobial Resistance (JPIAMR)-funded globally organized initiative of Traditional, Complementary and Integrative Healthcare (TCIH) and antimicrobial resistance (AMR)/infectious diseases research institutes, researchers in both human and veterinary medicine and global/regional policymakers. The aims of the initiative are:To develop a global “Traditional Solutions to Antimicrobial Resistance” network by mapping and connecting the research fields, research institutes, and researchers in human and animal healthcare and infrastructures involved in research on TCIH.To develop research agendas starting with at least one to three prioritized indications both in human and veterinary healthcare.To prepare grant proposals for research projects and the continuation of the network after the JPIAMR-funded GIFTS-AMR project.To communicate to relevant stakeholders the existence, activities, and output (e.g., research agendas, website) of the network, both online (report on website and webinars) and during an (online) international conference [69].

The GIFTS-AMR network involved seventeen research institutes and six additional organizations worldwide working in TCIH on AMR in human and/or animal healthcare. The TCIH systems are Anthroposophic Medicine, Ayurveda, Homeopathy, Phytotherapy/Western herbal medicine, and Traditional Chinese Medicine. A broad range of research types is covered among the network, with many working in clinical and pre-clinical research and, for example, on ethnomedicinal surveys, guideline development, and public health research.

### 1.6. Study Aim

This study aims:To present the research themes, priorities, and projects of the developed global TCIH research agenda on AMRTo describe its added value to the WHO Global Research Agenda for Antimicrobial Resistance in Human Health 2023 and the WHO/FAO/UNEP/WOAH One Health Priority Research Agenda for Antimicrobial Resistance 2023.

## 2. Results

### 2.1. The Global TCIH Research Agenda on AMR

With the use of an adapted version of the nominal group technique, based on consensus building at first in six separate groups and finally with all project members, the following six chapters on the main TCIH research themes were produced that provide the input for the research agenda and suggested advocacy actions related to this agenda:The value of TCIH medicinal productsThe best TCIH product–market combinationsThe most promising TCIH medicinal products for high-quality randomized controlled trials (RCTs)The use of limited evidence and real-world evidence of safety and effectivenessThe transition towards full integration of TCIH in the medical systemsIncreasing the accessibility of TCIH medicinal products for infections (information)

The chapters are summarized into 14 research priorities, with prioritized research projects for the next 10 years (Table 1). The global TCIH research agenda has been presented during the online Research Conference on TCIH Strategies for Infections and AMR Reduction (9–10 November 2023) and is published [70].

### 2.2. The Additions to the WHO Research Agenda on AMR 2023

The WHO research agenda on AMR 2023 contains 40 priorities in 13 AMR areas across five themes. The global TCIH research agenda contributes to three of the five themes and seven of the 13 related AMR areas of the WHO research agenda on AMR 2023 (Figure 2). The numbers between brackets in relation to the AMR areas refer to the research priority included in the themes. For example, the global TCIH research agenda adds to the AMR area “Water, sanitation, and hygiene” (WASH) of theme 1 Prevention, which contains research priorities 1 and 2 (see Table 1 for more detailed information).

### 2.3. The Additions to the WHO/FAO/UNEP/WOAH Research Agenda 2023

The global TCIH research agenda contributes to four of the five pillars and the related priority research areas of the WHO/FAO/UNEP/WOAH research agenda 2023. Figure 3 lists the four pillars with their related priority research areas (see for more detail Table 1).

### 2.4. Newly Defined Research Themes and Research Priorities of the Global TCIH Research Agenda in Addition to the Two Existing Global AMR Research Agendas

The global TCIH research agenda includes two research themes with four research priorities that have not previously been included in the existing research agendas. These newly defined research themes and priorities are:Patient preferences and stakeholders’ needs for non-antibiotic prevention and treatment strategies for infections
oAssess patients’/animal owners’/farmers’ preferences, use, satisfaction, and acceptability of TCIH MPs in LMICs and developed countries.Use of limited evidence and real-world evidence:oDevelop an adapted Evidence-to-Recommendation (EtR) system for TCIH MPs for infections using available evidence and additional arguments to weigh up the available information.oInvestigate the feasibility of using identified additional arguments in an existing EtR framework.oInvestigate the acceptability and need for improvements of these EtR procedures for all TCIH modalities in all countries.

In addition, the global TCIH research agenda includes three research priorities that add to already existing themes in these research agendas. These newly defined priorities for the two existing themes are:Safety, (cost-)effectiveness, benefits/risks ratios, and benefits/costs ratios of TCIH strategies in human and veterinary medicineoInvestigate the feasibility and acceptability of integrating traditional and complementary approaches with conventional primary healthcare (for humans and animals) as a strategy to support the delayed use of antibiotics.oInvestigate the types and working mechanisms of health promotion/resilience and antimicrobial effects of TCIH MPs.Implementation and information toolsoInvestigate the conceptual differences between conventional medicine and TCIH, which present a barrier to the acceptability and implementation of TCIH prevention and treatment of infections strategies.

## 3. Discussion

In 2023, two major global research agendas on AMR were published: (1) the “A One Health priority research agenda for antimicrobial resistance (2023). Research agenda by Food and Agriculture Organisation of the United Nations, United Nations environment programme, World Health Organization and World Organisation for Animal Health” and (2) the “Global research agenda for antimicrobial resistance in human health” (2023). In a JPIAMR-funded project (GIFTS-AMR), a global TCIH research agenda was developed based on a survey, interviews, and consensus meetings among the GIFTS-AMR network members as an adapted version of the nominal group technique [71]. The global TCIH research agenda adds to 19 of 40 research priorities of the WHO AMR research agenda 2023 and three of the five pillars of the WHO/FAO/UNEP/WOAH research agenda 2023. In addition, the TCIH research agenda adds two new research themes with four new research priorities and adds three new research priorities to already existing themes of the two global AMR research agendas.

### 3.1. Strength and Limitations

The first strength of this study is that it was based on the input and consensus building of 17 research institutes and non-research institutes in 17 countries, including both high-income and low–middle-income countries, and from the main Integrative Medicine modalities. The second strength is that its aims fit with and add to the aims of the two other global research agendas. A third strength is that the agenda is in line with the One Health approach. The main limitation is the use of a non-systematic approach for data collection, although this was outweighed by using an adapted version of the nominal group technique. The research agenda is limited by the fact that it is consensus-based instead of a systematic approach and researcher-based, because no other stakeholder groups were participating in the process. Although this was a transparent process, possible unconscious or unacknowledged biases on the part of the scientists or when it comes to applying the synthesis of scientific evidence to decisions on the research agenda may have occurred.

### 3.2. Future Perspective

Given the still existing threat of AMR, the rising use of antibiotics, the related costs, and the fact that “It is now outdated to state, as many medical doctors still do”, that “there is no evidence for herbal medicines” [72], and no evidence for homeopathic medicine [28], the developed TCIH research agenda can be used as an additional strategy to reduce AMR and the (inappropriate) use of antibiotics. Identified future actions supporting the implementation of the GIFTS-AMR global research agenda are:
Continue and broaden the international research network, building on the existing GIFTS-AMR network, consisting of TCIH and conventional researchers with different backgrounds and skills (veterinary/medical, human sciences, philosophical, historical, political, etc.) as well as professionals (doctors, veterinarians, pharmacists, biologists, physicists, chemists, pharmacologists, etc.) who are working towards a health-oriented healthcare system.Develop a formal, trustworthy global scientific ‘committee/working group’, recognized by conventional and TCIH stakeholders, which provides valid information on TCIH research, education, and information tools for the prevention and treatment of infections and reduction in AMR. This committee/working group should be responsible for the development of specific, high-quality databases on TCIH strategies and scientific evidence of TCIH research in this field to ensure patients, animal owners, farmers, healthcare professionals, and other stakeholders can access user-friendly evidence-based information/advice on TCIH.Develop and promote databases such as CAM on PubMed^®^ and VHL by PAHO and CABSIN among professionals and academic researchers.Develop and use standards for evidence-based education of healthcare professionals (human and veterinary)—undergraduate and postgraduate.Consider a broad range of sources (e.g., context-based/real-world evidence, users’ preferences) in research and guideline development.Promote guideline development considering both quantitative and qualitative research, including results of ‘real-world evidence’ studies.Expand the role of ‘patient choice’ in future research, guideline development, and education (PPI, public and patient involvement) in those countries where this is not, or insufficiently, organized.Prioritize research on health/resilience promotion rather than disease control only.Develop information tools (eHealth, website) to provide easily accessible information on evidence.Promote One Health research and the collaboration between conventional medicine and TCIH in human, animal, and plant sectors at regional, national, and international levels in a timely manner while preserving the integrity of prescribing individualized TCIH treatments and while considering research on health/resilience promotion and disease-specific prevention rather than disease control only.Promote publicly funded research on TCIH treatments, their efficacy/(cost-)effectiveness and safety, and their underlying mechanisms or modes of action.

## 4. Methods

### 4.1. Research Questions

What are the research themes, priorities, and prioritized projects of the global TCIH research agenda on AMR for the next 10 years?What does the global TCIH research agenda add to the forty research priorities, the thirteen AMR areas across five themes of the WHO AMR, and the five pillars of the WHO/FAO/UNEP/WOAH research agendas 2023?Which newly defined research themes and priorities can be added to the two global AMR research agendas?

### 4.2. Methods

#### 4.2.1. Data Collection

The data collection was executed in four phases. The methodology used in Phases 1, 3, and 4 was an adapted version of the nominal group technique [71] to build consensus among experts.

In the first phase, members of the GIFTS-AMR project were interviewed and a survey was disseminated among them (17 research institutes in 17 countries (see for the full list Appendix A)). These activities aimed to collect expertise-based information and opinions on the (possible) contributions of TCIH on the prevention and treatment of human and animal infections and reducing AMR and to collect input for the research agendas from global experts in the field of TCIH, infectious diseases, and AMR.

In the second phase, based on content-wise clustering of themes within the collected information, the first objectives of the research agenda were identified, and a first draft of research agenda topics and research questions per topic were formulated and shared within the network. Members were invited to add and/or change topics.

In Phase 3, for each main topic, a working group was formed to work on describing the background, the main topics for the development of the research agenda, the current situation, the research themes, and the prioritized research projects for the next ten years.

Consensus on the research agenda among the GIFTS-AMR group members was reached in Phase 4, following several rounds of providing input and comments on concept documents.

#### 4.2.2. A Comparison with the Two 2023 AMR Research Agendas

Finally, the topics of the TCIH research agenda were compared with the forty research priorities and the thirteen AMR areas across five themes of the WHO AMR research agenda 2023 and the five pillars of the WHO/FAO/UNEP/WOAH research agenda 2023 to identify (a) additional research themes and research priorities of the TCIH research agenda to these existing research priorities and pillars, and (b) newly defined research themes and research priorities in addition to the two global AMR research agendas 2023.

## Figures and Tables

**Figure 1 antibiotics-14-00102-f001:**
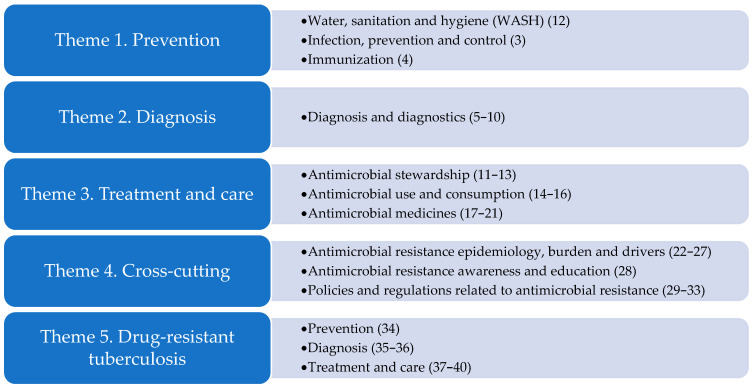
The five themes and thirteen related AMR areas of the WHO research agenda.

**Figure 2 antibiotics-14-00102-f002:**
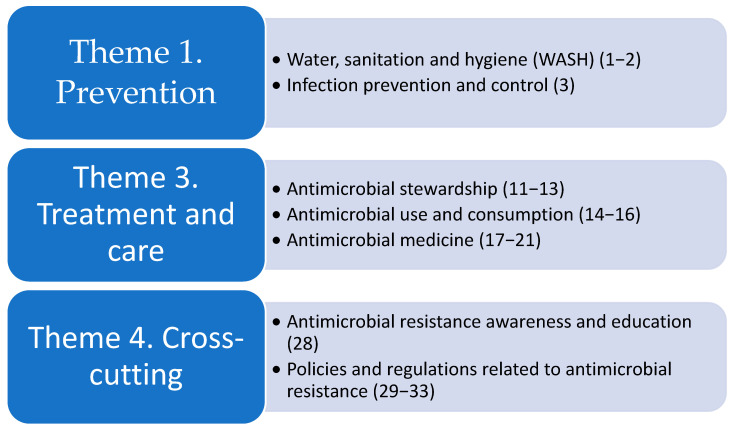
The additions of the global TCIH research agenda to the WHO research agenda on AMR 2023.

**Figure 3 antibiotics-14-00102-f003:**
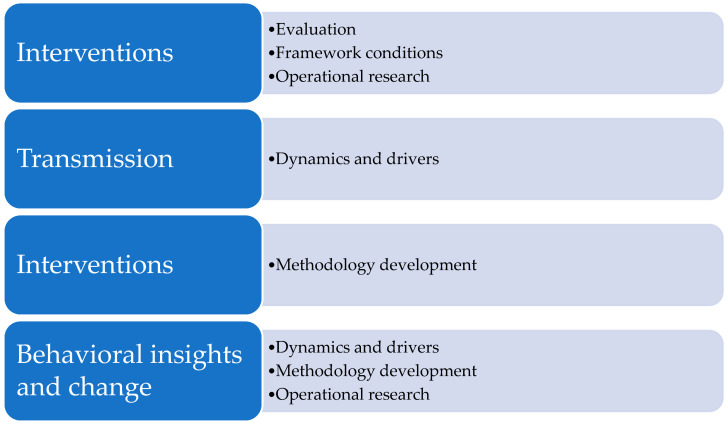
The additions of the global TCIH research agenda to the WHO/FAO/UNEP/WOAH research agenda 2023.

**Table 1 antibiotics-14-00102-t001:** TCIH research themes, research priorities, prioritized research projects for the next 10 years, and contributions to two global AMR research agendas.

Research Themes	Research Priorities	Prioritized Research Projects for the Next 10 Years	Contribution to the WHO Global Research Agenda (2023) *	Contribution to the WHO/FAO/UNEP/WOAH Global Research Agenda (2023) **
Patient preferences and stakeholders’ needs for non-antibiotic prevention and treatment strategies for infections	Assess patients’/animal owners’/farmers’ preferences, use, satisfaction, and acceptability of TCIH MPs in LMICs and developed countries.	Map out the qualitative and quantitative studies on patients’/animal owners’/farmers’ preferences, use, satisfaction, and acceptability of TCIH MPs as alternatives to antimicrobials (scoping review).	New ***	New ***
Safety, (cost-)effectiveness, benefits/risks ratios, and benefits/costs ratios of TCIH strategies in human and veterinary medicine	2. Investigate the safety, working mechanisms, and efficacy/(cost-)effectiveness of the most promising TCIH MPs for indications where antimicrobials are commonly over-used.	Investigate the (cost-)effectiveness of the most promising TCIH MPs for acute, uncomplicated URTI and rUTI symptom control and reduction in antibiotics use in primary care and hospital ER departments. e.g., investigate the effectiveness of Traditional Chinese Medicine compared with standard conventional treatment and/or placebo for the prevention of UTIs in patients with a history of rUTIs. Investigate the (cost-)effectiveness of FeverApp/FeverFriend tools for fever management on symptom control and reduction in antimicrobials use in GP practices and hospital ER departments. Investigate the (cost-)effectiveness of the most promising TCIH MPs for uncomplicated diarrhea and RTIs in animals. e.g., investigate the effectiveness of homeopathy, compared with placebo, for the prevention of ETEC-related post-weaning diarrhea in piglets at risk. Systematically review the safety and effectiveness of TCIH MPs for gastrointestinal infections in humans and RTIs and gastrointestinal infections in animals.	Treatment and care—Antimicrobial stewardship (11–13)	Interventions—Evaluation
	3. Investigate the feasibility and acceptability of integrating traditional and complementary approaches with conventional primary healthcare (for humans and animals) as a strategy to support the delayed use of antibiotics.	Investigate the feasibility and acceptability of promising TCIH MPs for acute, uncomplicated URTIs and rUTIs in humans and uncomplicated diarrhea and RTIs in animals. Assess the acceptability and effectiveness of updated guidelines on the management of self-limiting infections, including recommendations to use TCIH to support delayed use of antibiotics.	New *** Treatment and care—Antimicrobial stewardship (11–13)	New *** Interventions—Evaluation
	4. Investigate the types and working mechanisms of health promotion/resilience and antimicrobial effects of TCIH MPs.	Investigate the types and working mechanisms of health/resilience promotion and antimicrobial effects of 3–5 TCIH MPs with moderate to high-quality evidence of effectiveness in clinical trials for acute, uncomplicated URTIs and rUTIs in humans and for uncomplicated diarrhea and RTIs in animals.	New ***	New ***
	5. Investigate the benefits/risks ratios and the benefits/costs ratios of TCIH MPs vs. antimicrobials for humans, animals, and the environment.	Investigate the effectiveness and reduction in adverse effects on animals and environments for TCIH MPs with moderate to high-quality evidence of effectiveness for uncomplicated URTIs and rUTIs in humans and for the prevention of ETEC-related post-weaning diarrhea in piglets at risk.	Prevention—Water, sanitation, and hygiene (WASH) (1–2) Prevention—Infection prevention and control (3) Treatment and care—Antimicrobial stewardship (11–13)	Transmission—Dynamics and drivers Interventions—Framework conditions
	6. Investigate the effects of whole system approaches (e.g., organic/biodynamic agriculture and TCIH whole medical system prevention and treatment) on a sustainable reduction in antimicrobial use and consumption.	Map out the qualitative and quantitative studies on the effects of whole medical systems and whole system approaches in (organic and biodynamic) farming on reducing antimicrobial use and consumption (scoping review).	Treatment and care—Antimicrobial use and consumption (14–16)	Interventions—Operational research
	7. Develop and evaluate valid score systems which weigh up relevant factors to identify and prioritize the most promising TCIH MPs for urgent indications which can be tested in high-quality RCTs, and which are usable and acceptable for relevant stakeholders.	Develop valid score systems that weigh up relevant factors to identify and prioritize the most promising TCIH MPs, which can be tested in high-quality RCTs for the treatment of acute, uncomplicated URTIs and rUTIs in humans and uncomplicated diarrhea and RTIs in animals in countries with major over-use of antimicrobials for these indications.	Treatment and care—Antimicrobial medicine (17–21)	Interventions—Methodology development
Use of limited evidence and real-world evidence	8. Develop an adapted Evidence-to-recommendation (EtR) system for TCIH MPs for infections using available evidence and additional arguments to weigh up the available information.	Develop and investigate the feasibility and acceptability of an adapted EtR system for TCIH MPs for the treatment of acute, uncomplicated URTIs and rUTIs in humans and for uncomplicated diarrhea and RTIs in animals in countries with major over-use of antimicrobials for these indications.	New ***	New ***
	9. Investigate the feasibility of using identified additional arguments in an existing EtR framework.		New ***	New ***
	10. Investigate the acceptability and need for improvements of these EtR procedures for all TCIH modalities in all countries.		New ***	New ***
Implementation and information tools	11. Investigate the conceptual differences between conventional medicine and TCIH, which present a barrier to the acceptability and implementation of TCIH prevention and treatment of infections strategies.	Build a combined expertise-and evidence-based theoretical model of TCIH treatment of acute, uncomplicated URTIs and rUTIs in humans and for uncomplicated diarrhea and RTIs in animals to overcome the barrier for acceptability and implementation while preserving the integrity of TCIH prescribing of individualized and non-individualized TCIH treatments, and while considering research on health/resilience promotion (realist review of complex interventions).	New ***	New ***
	12. Investigate the reasons for current guideline developers to decide on the (non-)inclusion of TCIH MPs in guidelines for the prevention and treatment of infections.	Investigate the reasons for the non-inclusion of TCIH MPs for infections in European countries for those TCIH MPs that already have an EMA status of Traditional use or Well-established use and/or are included in conventional guidelines (e.g., in the UK or Germany): EMA status o Ivy for coughs and common cold o Pelargonium sidoides for the common cold o German guidelines o Pelargonium sidoides for coughs (DEGAM Leitlinie Nr 11), rhinosinusitis (S2k-Leitlinie) o Thyme/Primrose for coughs (DEGAM Leitlinie Nr 11) UK guidelines o *Pelargonium sidoides* for coughs (NICE Cough (acute) guideline)	Treatment and care—Antimicrobial stewardship (11)	Behavioral insights and change—Dynamics and drivers
	13. Develop and evaluate information tools (websites, eHealth) to provide easily accessible and trustworthy advice for patients on TCIH strategies for self-management of common infections in which antimicrobials are commonly over-used, and trustworthy information for clinicians (including evidence of safety, (cost-)effectiveness, use in clinical practices); and, additionally, on benefits/risks ratios and benefits/costs ratios for research and policy-making.	Develop and evaluate the usability and acceptability of TCIH MPs for URTIs prototype app for use in different countries (language and cultural adaptation). Develop the app further for rUTIs in humans and diarrhea in animals. Implement and adapt (language and cultural) the FeverApp and FeverFriend app for humans with over-use of antimicrobials related to fever management, in countries other than Germany and The Netherlands. Develop and evaluate a FeverApp/FeverFriend app for use in veterinary medicine. Develop and evaluate the quality, usability, and acceptability of a TCIH website with science-based information on TCIH strategies for the prevention and treatment of infections for research, education, and use in clinical practices.	Cross-cutting—Antimicrobial resistance awareness and education (28)	Interventions—Operational research Behavioral insights and change—Methodology development
	14. Implement TCIH prevention and treatment strategies as part of a One Health approach in relevant antimicrobial stewardship programs (ASPs).	Investigate barriers and promotors of the implementation of TCIH prevention and treatment strategies as part of a One Health approach in relevant ASPs. Investigate implementation methods that will enable TCIH prevention and treatment strategies as part of a One Health approach, including the collaboration between conventional medicine and TCIH in a timely manner, in relevant ASPs.	Cross-cutting—Policies and regulations related to antimicrobial resistance (29–33)	Interventions—Methodology development Behavioral insights and change—Operational research

DEGAM: Deutsche Gesellschaft für Allgemeinmedizin und Familienmedizin; ETEC: Enterotoxigenic *Escherichia coli*; GP: general practitioner; NICE: National Institute for Health and Care Excellence. * The list refers to the broader research topics and the related research priorities (numbers) for AMR in the WHO agenda: https://cdn.who.int/media/docs/default-source/antimicrobial-resistance/amr-spc-npm/who-global-research-agenda-for-amr-in-human-health---policy-brief.pdf?sfvrsn=f86aa073_4&download=true (accessed on 14 January 2025) ** The list refers to the priority research (sub)areas in the UN/WHO/WOAH agenda https://www.fao.org/3/cc6213en/cc6213en.pdf (accessed on 14 January 2025) *** This item is new to the content of the existing research agenda. They have not previously been included in the existing research agendas and thus add new fields/topics of research to them.

## Data Availability

The whole research report is Open Access and available at: https://louisbolk.nl/media/pdf/Research-Agenda.pdf (accessed on 14 January 2025).

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
