# Peer review of "The Introduction of the Global Traditional, Complementary, and Integrative Healthcare (TCIH) Research Agenda on Antimicrobial Resistance and Its Added Value to the WHO and the WHO/FAO/UNEP/WOAH 2023 Research Agendas on Antimicrobial Resistance"

_antibiotics, 2025, doi:10.3390/antibiotics14010102_

Round 1

Reviewer 1 Report

Comments and Suggestions for Authors

The article by Baars and colleagues (Manuscript ID: antibiotics-3409388) presents a core of the global Traditional, Complementary &  Integrative Healthcare research agenda on antimicrobial resistance and its added value to the other projects with the same scope. The article is of medical and pharmacological relevance for both human and veterinary medicine. It is well-designed and informative. However, some figures or diagrams instead of the textual explanations of the: 1. five themes and related antimicrobial resistance areas (rows 122-140);  2. the additions to the WHO research agenda on AMR 2023 (rows 391-400); 3. the list of four pillars with their related priority research areas (rows 406-418) can bring additional quality. In a row 83 authors cite the European Antimicrobial Resistance Surveillance Network data from 2018. Maybe there are some changes in the latest report (https://www.ecdc.europa.eu/sites/default/files/documents/antimicrobial-resistance-ECDC-WHO-executive-summary-2023-data.pdf).

Minor observations:

Row 17 – rephrase the “urgency of AMR”

Rows 22 – 23 and 26-28 – “research agenda or agendas” three times in a sentence, it should be rephrased

Row 29 – WHO (abbreviation introduced without explanation)

Row 30 – WHO/FAO/UNEP/WOAH (abbreviation introduced without explanation)

Row 42 – UK (abbreviation introduced without explanation)

Row 45 – US $ (abbreviation introduced without explanation)

Rows 48-49 – “control of resistant bacteria” two times in a sentence, it should be rephrased

Row 60 – ECDC (abbreviation introduced without explanation)

Row 102 – OECD (abbreviation introduced without explanation)

Row 179 – “uncomplicated respiratory tract infections (RTIs)” – URTI should be in a brackets

Row 205 – MP (abbreviation introduced without explanation)

Rows 209-210 – AT, DE, NL, USA (abbreviations introduced without explanation)

Rows 221-222 – “Thyme/Primrose for coughs [30], and Pelargonium sidoides for coughs [30]” should be rephrased with Thyme/Primrose and Pelargonium sidoides for coughs [30]”

Row 224 – EMA (abbreviation introduced without explanation)

Row 235 – RCTs (abbreviation introduced without explanation)

Rows 282 and 285 – “(e.g.” – sentences are not finished properly

Row 329 - “research” three times in a sentence, it should be rephrased

Rows 331-333 - “research agenda” three times in a sentence, it should be rephrased

Rows 334-335 - “research” three times in a sentence, it should be rephrased

Row 342 – “interviews were organized with” - it should be rephrased

Row 448 – Table 1: Some abbreviations have explanation: “evidence-to-recommendation (EtR)”; “antimicrobial stewardship programs (ASPs)” and others don't (ER, GP, ETEC, RCTs, EMA, NICE). It would be more clear if the table contains only abbreviations, and below the table there is a legend with their explanations. Also “*** This item is new to the content of the existing

research agenda” is not well explained (in what sense, what can we expect and in which fields)

Row 585 - reference with wrong formatting and bacterial species and genes should be in italics

Row 607 - reference with wrong formatting

Row 650 and 659 - bacterial species should be in italics

Author Response

Cover letter resubmission revised article Antibiotics-3409388

First of all we would like to thank the reviewers for their quick and constructive feedback. Hereby you will find the revisions we have made and the comments to the reviewers.

Reviewer 1

  • We have put the three texts parts in three figures as suggested in the manuscript.
  • We have changed the text on row 83 according to the suggested ECDC-WHO document.
  • All but two “minor observations” have been changes as requested in the text.
    • However, “US$” (Row 45) is a quote that therefore cannot be changed in the text. Therefore we added Abbreviations (row 501 on p. 17).
    • Row 179 – “uncomplicated respiratory tract infections (RTIs)” – URTI should be in a brackets: This is correct. URTIs refer to upper respiratory tract infections. We therefore changed URTIs into upper RTIs

On behalf of all authors, sincerely,

Erik Baars

Reviewer 2 Report

Comments and Suggestions for Authors

Line 51-52: "development of new antibiotics and the role of artificial intelligence (AI)" seems confusing, do the authors actually mean development of new antibiotics with the application of artificial intelligence?

Line 83-94: Text font for these sentences is different than other sentences.

Line 110: Wrong reference number.

Line 470-471, 559, 564, 585-586, 590-591, 598, ... : Different text font.

Author Response

Cover letter resubmission revised article Antibiotics-3409388
First of all we would like to thank the reviewers for their quick and constructive feedback. Hereby you will find the revisions we have made and the comments to the reviewers.

     All “minor observations” have been changed as requested in the text.

On behalf of all authors, sincerely,
Erik Baars

Reviewer 3 Report

Comments and Suggestions for Authors

It should be highlighted that this is a review article and not a research article. The manuscript requires extensive corrections to the formatting to make it presentable. The current state of formatting makes it a very tedious read. 

Author Response

Cover letter resubmission revised article Antibiotics-3409388
First of all we would like to thank the reviewers for their quick and constructive feedback. Hereby you will find the revisions we have made and the comments to the reviewers.

  • “It should be highlighted that this is a review article and not a research article”:
    • We do not agree with the reviewer that it is a review article instead of a research article. As described in the Methods section, an adapted version of the nominal group technique [70] was used to build consensus among experts. In addition was there a comparison with the two other global AMR 2023 agendas. Both elements are descriptive research.
  • “The manuscript requires extensive corrections to the formatting to make it presentable”:
    • We cannot change this because this is the task of the journal. Nevertheless, we would be glad to adapt the structure of the text in case of more specific instructions. Otherwise we hope the journal will take care of this.

On behalf of all authors, sincerely,
Erik Baars